# Smoothness Evaluation Indices during Sit-to-Stand-to-Sit Motions in Healthy Older Females and after Hip Fracture Using an Accelerometer: A Pilot Study

**DOI:** 10.3390/geriatrics8050098

**Published:** 2023-10-01

**Authors:** Takeshi Shimamura, Hitoshi Ishikawa, Hiromi Fujii, Hiroshi Katoh

**Affiliations:** 1Department of Rehabilitation, Kumamoto Health Science University, 325 Izumi-machi, Kita-ku, Kumamoto 861-5598, Japan; 2Graduate School of Health Sciences, Yamagata Prefectural University of Health Sciences, 260 Kamiyanagi, Yamagata 990-2212, Japan

**Keywords:** sit-to-stand-to-sit motion, smoothness, accelerometer, harmonic ratio, power spectrum entropy

## Abstract

Background: Studies that quantify the quality of sit-to-stand-to-sit (STS) motions, particularly in terms of smoothness, are limited. Thus, this study aimed to investigate the possibility and usefulness of quality evaluation during STS motions. Methods: This cross-sectional study enrolled 36 females aged >60 years, including 18 females each in the healthy and hip fracture groups. Measurements were performed at two different speeds: five STS as fast as possible (STSF) and two seconds for each motion (STS2s). Indices of smoothness, including harmonic ratio (HR) and power spectrum entropy (PSE), were calculated and compared from the measured data in each of the three axial directions. Results: HR in the vertical direction was significantly higher in the healthy group (STSF: 3.65 ± 1.74, STS2s: 3.42 ± 1.54) than in the hip fracture group (STSF: 2.67 ± 1.01, STS2s: 2.58 ± 0.83) for STSF and STS2s. Furthermore, PSE for all directions and triaxial composites were significantly lower for STS2s (the healthy group (mediolateral (ML): 7.63 ± 0.31, vertical (VT): 7.46 ± 0.22, anterior–posterior (AP): 7.47 ± 0.15, triaxial: 7.45 ± 0.25), the hip fracture group (ML: 7.82 ± 0.16, VT: 7.63 ± 0.16, AP: 7.61 ± 0.17, triaxial: 7.66 ± 0.17)). Conclusions: This study suggests the usefulness of HR and PSE as quality evaluations for STS motions.

## 1. Introduction

Motion analysis is essential for rehabilitation to improve the basic actions of sit-to-stand-to-sit (STS) and gait motions. Several motion analyses of gait motion have been explored in the field of rehabilitation [1]. Gait analysis can be performed using various methods, including quantitative measures such as speed, stride length, and gait ratio [2] as well as the evaluation of motion quality by analyzing acceleration. In recent years, studies using accelerometers have been increasing in clinical settings as it is a simple evaluation tool with few limitations on measurement [3]. In particular, harmonic ratio (HR) and power spectrum entropy (PSE) have been used as indices of motion quality in previous studies focusing on smoothness during gait. HR evaluates the regularity, harmony, and symmetry of behavior [4,5,6], and stable behavior has been reported to be associated with higher values. PSE evaluates smoothness based on the probability of superimposing higher frequencies [7,8,9], and it has been reported that good behavior is associated with lower values. Particularly in older adults, the PSE of maximum speed gait is less correlated with physical functions such as grip strength; however, a distinction has been made between those with and without a history of falls [7]. These smoothness indices are effective in providing information that is not provided by quantitative indices [7,10], thus indicating their usefulness in clinical settings.

Conversely, STS is a pre- and post-gait motion essential for smooth activities of daily living (ADL). The 30 s standing test using frequency and the 5 sit-to-stand test (5STS) measuring motion time are widely used to assess STS ability, and the association between motion time and worsening ADL in older adults has been reported [11]. Additionally, STS ability has been associated with maximum gait speed [12], frailty and gait speed, and ADL and quality of life disability in females >60 years old [13]. Furthermore, Bagalà et al. [14] used accelerometers to measure variations in acceleration during lie-to-sit-to-stand-to-walk movements and reported their usefulness in assessing the quality of performance. Zablotny et al. [15] reported that observational analysis of sit-to-stand motion in older adults after hip fracture can be supported with quantitative data to evaluate intervention. Other researchers used accelerometers to evaluate the time and speed of mobility [16], whereas some examined the amount of physical activity after fracture [17]. Although many studies have evaluated and examined movement qualities such as smoothness [5,7,14,18], to the best of our knowledge, none of the studies have used smoothness indices in STS motion. Therefore, this study aimed to investigate the differences in the smoothness of motion during STS in healthy older adults and postoperative patients with hip fractures using quality indices. In this study, it was hypothesized that the healthy group had a higher quality of movement, higher HR, and lower PSE than the hip fracture group.

## 2. Materials and Methods

### 2.1. Study Design

The present study is a cross-sectional study.

### 2.2. Participants

The participants included 18 healthy women aged ≥60 years without psychiatric, neurological, or musculoskeletal diseases and without any history of diseases affecting movement (healthy group) and 18 women with hip fractures with Mini-Mental State Exam scores of ≥24 without any history of previous dementia who were able to perform the movement and consented to participate in the study (hip fracture group) (Figure 1). This study was approved by the Ethics Committee of Kumamoto Kenhoku Hospital (2020-012). Data collection was conducted 22 months after obtaining ethics approval.

### 2.3. Ethical Procedures

All participants were informed about the purpose of the study as per the Declaration of Helsinki, and their informed consent was obtained before starting the study.

### 2.4. Measurements

The 9-axis wireless motion sensor SS-MS-HMA16G15 (Motion Sensor) (Sports Sensing Inc., Fukuoka, Japan) was used for measurement. It was 53, 38, and 11 mm in length, width, and thickness, respectively. The 9 axes comprised 3-axis acceleration (16 G), 3-axis angular velocity (1500 dps), and 3-axis geomagnetic sensor (10 gauss). Acceleration was noted in 3 axes: mediolateral (ML), vertical (VT), and anterior–posterior (AP) directions. The motion sensor was attached to the participant’s third lumbar spinous process (L3) using an elastic band, following the method of Moe Nilssen et al. (Figure 2) [19]. The sampling frequency was set at 100 Hz. The task motion included five STS repetitions at two different speeds and one maximum speed gait. The STS measurements began with the participants sitting in a chair without a backrest or armrests, with their arms crossed on their chests [20,21,22]. Seat height was adjusted using adjustable chairs, hip and knee joints were set at approximately 90° [22], and the distance between the right and left feet was the width of the superior anterior iliac spine [20]. The first task motion consisted of five STS fast (STSF) repetitions at maximal speed, whereas the second was performed using a metronome to control sit-to-stand and stand-to-sit for 2 s (STS2s) each [23] for five repetitions. Measurements were taken after practice with a 1 min rest period. Gait motion was measured for a 10 m steady-state gait with 3 m aided paths in front and behind. The instructions for the maximum speed gait were “please gait as fast as you can” [12]. Rest intervals of ≥1 min were provided to verbally confirm that there was no fatigue before proceeding to the next measurement. Measurements of the hip fracture group were estimated for 7 weeks postoperatively and were performed within 3 days of the measurement being possible.

### 2.5. Data Processing

Analysis parameters included 5STS, HR, and PSE, and the acceleration data were analyzed after low-pass filtering using a Butterworth filter of 20 Hz [5]. STS motion analysis is affected by the tilt of the motion sensor during motion in the 3-axis acceleration data and by gravity due to the difference between relative and absolute coordinates. Thus, the VT and AP directions, in which the effects of gravity are greater, were corrected [23]. STS detection was determined at the point at which the angular velocity of the frontal axis during motion switched from negative to positive after two repetitions of acceleration and deceleration (Figure 3). To calculate the indices, the start of motion was defined as the point at which the angular velocity of the frontal axis during motion exceeded three times the standard deviation of the mean angular velocity in the sitting position. In the gait motion, the walking cycle was extracted by referring to the peak value before the change in AP acceleration from positive to negative with heel contact occurred (Figure 3) [24]. Triaxial composite direction analysis was conducted using the square root of the sum of the squared accelerations in the three directions.

#### 2.5.1. Five Sit-to-Stand Test

5STS was calculated from STSF and was defined as the time from the onset of motion to the end of the fifth standing.

#### 2.5.2. Harmonic Ratio

HR in the STS motion was analyzed as one cycle for each of the sit-to-stand and stand-to-sit motions owing to the fact that the directions of these motions are different. The discrete Fourier transform and the even/odd of the first twenty harmonic coefficients [5,6,25], two sit-to-stand, and two stand-to-sit were used to calculate HR (Figure 4). Representative values include the average of the two values calculated 2–3 and 3–4 times.

#### 2.5.3. Power Spectrum Entropy

Fast Fourier transform with Hanning’s window function and normalized time series spectrum after excluding the first component were used to calculate PSE (Figure 4) [7,8,9].

### 2.6. Statistical Analyses

The normality of the data was assessed using the Shapiro–Wilk test, and equal variance was assessed using the Levene test. Moreover, the two-sample *t*-test and Mann–Whitney U test were used to compare the participants in the healthy and hip fracture groups. Effect size (ES) was calculated in terms of Cohen’s d, where d = 0.2, 0.5, and 0.8 representing small, medium, and large effect sizes, respectively [26]. Notably, *p*-values of <0.05 were considered statistically significant. R4.2.1 (CRAN, freeware) for Windows was used to perform all statistical analyses. During the comparison of 5STS and maximum gait speed (MGS), analysis of covariance adjusted for age and weight was used to evaluate the results. Further, the sample size was calculated using G*Power 3.1.9.7 (Heinrich Heine University, Dusseldorf). The significance level was set at α = 0.05, power was set at 1 − β = 0.8, ES (d) of 0.988 was assumed in the preliminary study, and the minimum number of participants in the group was recommended to be 18.

## 3. Results

### 3.1. Characterization of the Sample and Comparison of the Two Study Subgroups

Age was significantly lower in the healthy group (63.1 ± 3.2 years) than in the hip fracture group (76.2 ± 7.8 years), but weight and body mass index (BMI) were significantly higher in the healthy group participants (weight: 52.6 ± 9.0 kg, BMI: 22.1 ± 3.1 kg/m^2^) than in the hip fracture group participants (weight: 45.3 ± 5.7 kg, BMI: 20.0 ± 2.4 kg/m^2^) (Table 1).

### 3.2. Five Sit-to-Stand Test Compared with Maximum Gait Speed

The time required for 5STS was significantly lower in the healthy group (8.75 ± 1.05 s) than in the hip fracture group (12.17 ± 3.20 s) (*p* < 0.001, ES = 1.43). The MGS was significantly higher in the healthy group (1.93 ± 0.21 s) than in the hip fracture group (1.09 ± 0.27 s) (*p* < 0.001, ES = 3.51). Notably, the analysis of covariance with age and BMI as covariates revealed that MGS was significantly higher in the healthy group than in the hip fracture group (*p* = 0.012) (Table 2).

### 3.3. Harmonic Ratio Comparison

The HR for STSF motion in the VT direction was significantly higher in the healthy group (3.65 ± 1.74) than in the hip fracture group (2.67 ± 1.01) (*p* = 0.047, ES = 0.69); moreover, HR for STS2s motion in the VT direction was significantly higher in the healthy group (3.42 ± 1.54) than in the hip fracture group (2.58 ± 0.83) (*p* = 0.049, ES = 0.69) (Table 3).

### 3.4. Power Spectrum Entropy Comparison

STSF analyzed 512 intermediate movements (5.12 s), whereas STS2s analyzed 2048 intermediate movements (20.48 s). The PSE for STSF motion was significantly higher in the healthy group (AP: 5.78 ± 0.16, triaxial: 5.85 ± 0.21) than in the hip fracture group (AP: 5.61 ± 0.16, triaxial: 5.68 ± 0.18) in the AP (*p* = 0.002, ES = 1.10) and triaxial composite (*p* = 0.015, ES = 0.85) directions. Meanwhile, the PSE for STS2s motion was significantly lower in the healthy group (ML: 7.63 ± 0.31, VT: 7.46 ± 0.22, AP: 7.47 ± 0.15, triaxial: 7.45 ± 0.25) than in the hip fracture group (ML: 7.82 ± 0.16, VT: 7.63 ± 0.16, AP: 7.61 ± 0.17, triaxial: 7.66 ± 0.17) in the ML (*p* = 0.035, ES = 0.74), VT (*p* = 0.010, ES = 0.88), AP (*p* = 0.014, ES = 0.86), and triaxial composite (*p* = 0.003, ES = 0.97) directions (Table 4).

## 4. Discussion

In the present study, the average time for 5STS was 8.75 s in the healthy group. In previous studies on healthy participants by age, it has been reported that the average time taken for 5STS was 7.8, 9.3, and 10.8 s for participants in their 60s, 70s, and 80s [27], and the required time tended to be longer in older age groups. The values for healthy participants in the present study are considered to be reasonable when compared with the values reported in other studies [28,29]. By contrast, in the hip fracture group, the 5STS of three participants in their 60s, nine in their 70s, and five in their 80s were 9.7 ± 0.9, 12.0 ± 2.9, and 12.5 ± 2.8 s, respectively. The average 5STS of the hip fracture group was 12.17 s, suggesting that the participants of this group took a significantly longer time than those of the healthy group of the same age in previous studies.

Further, the mean MGS was 1.93 m/s and 1.09 m/s in the healthy and hip fracture groups, respectively, indicating lower speed in the hip fracture group than in the healthy group. Regarding previous studies, the MGS of healthy females aged ≥60 years was 1.9 m/s [12] and that of females aged ≥70 years was 1.21 m/s [30]. These results suggest that patients in the hip trauma group are more likely to have decreased STS and gait performance than healthy older adults in the same age group.

HR has been analyzed for two cycles with one step as one cycle in previous studies on gait motion, and the more the value increases, the more harmonic and smooth the motion [4,6]. This study attempted to analyze one sit-to-stand and one stand-to-sit cycle as one cycle and revealed a significantly higher HR during STS in the healthy group than in the hip fracture group in both STSF and STS2s in the VT direction. Thus, HR demonstrated the potential of evaluating the smoothness of STS. According to Pai et al., the increase in momentum during controlled sit-to-stand motion was smaller in the AP direction than in the VT direction due to changes in velocity, suggesting a difference in the strategy [31]. Therefore, the STS is primarily in the direction of gravity (antigravity), suggesting its usefulness in evaluating the main direction of motion. Additionally, the ESs for both STSF and STS2s were medium in the VT and triaxial composite directions, suggesting the usefulness of the triaxial evaluation.

PSE was reported to be smooth at low values and otherwise at high values [8]. The PSE for STSF was higher in the healthy group, indicating decreased motion smoothness in the healthy group. A previous study on gait motion reported PSE as independent of motion speed [7]; however, the analysis characteristics revealed the necessity to have an equal number of data for the comparison. Additionally, they reported that PSE is applied to movements for which time series spectra are obtained, but cyclic movements are desired [8]. Each cycle takes >1.5 s during STS, which is longer than gait, corresponding to 1–3 cycles in the 5.12 s analysis. Therefore, the different motion speeds in STSF may affect the value because of the different number of cycles included in the analysis.

By contrast, the STS2s evaluation aligned motion speed with the number of analysis cycles, and PSE was significantly lower in the healthy group for all directions and triaxial composites, suggesting the usefulness of quality evaluation. Furthermore, the ES was medium in the ML direction and large in the VT, AP, and triaxial composite directions, indicating the potential of the evaluation of ES in the triaxial composite direction by specifying the motion. Thus, PSE for STSF is difficult to compare between groups with remarkably different maximum speeds, and comparisons may be possible if STS2s motions were feasible. Hence, the evaluation of PSE for STS between groups requires the specification of the motion, and smoothness can be evaluated by aligning the amount of analysis data. The contents of the motion specifications and analysis methods for future tasks should be considered for STS2s evaluation for persons unable to perform it due to the large load.

### Limitations

This study has some limitations. First, it was not possible to exclude the effects of age and BMI in healthy older women (≥60 years) and in patients after hip surgery. Future comparisons with healthy groups including age and BMI are warranted. Second, it was very difficult to generalize the results of our study because the characteristics of men were not identified in the study. Third, because a 20 Hz low-pass filter was used during data processing, the effect of higher frequencies remains unclear. Fourth, the measurements were performed in the period of preparation for hospital discharge, which may have affected the results owing to anxiety about movement among the participants. Further studies to examine the ease of assessment and the amount of exercise load as well as the abovementioned factors are needed to utilize the results of our study for effective rehabilitation.

## 5. Conclusions

The present study examined the possibility of evaluating quality indicators for sit-to-stand-to-sit. Harmonic ratio and power spectrum entropy for sit-to-stand-to-sit were rarely reported in previous studies and have a higher motion load than gait, but they provide a new perspective on evaluation when determined at the appropriate time. The evaluation of the smoothness of motion at first is useful by evaluating harmonic ratio for sit-to-stand-to-sit fast, followed by power spectrum entropy and harmonic ratio for sit-to-stand and stand-to-sit for 2 s. The addition of these quality evaluations to the quantitative evaluation would assist in more appropriate motion evaluation and intervention consideration.

## Figures and Tables

**Figure 1 geriatrics-08-00098-f001:**
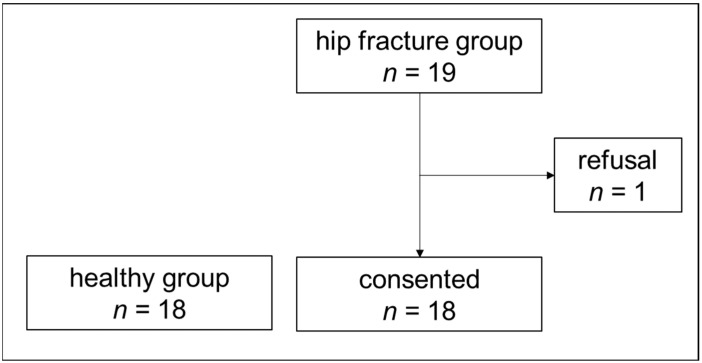
Recruitment of the healthy group and hip fracture group.

**Figure 2 geriatrics-08-00098-f002:**
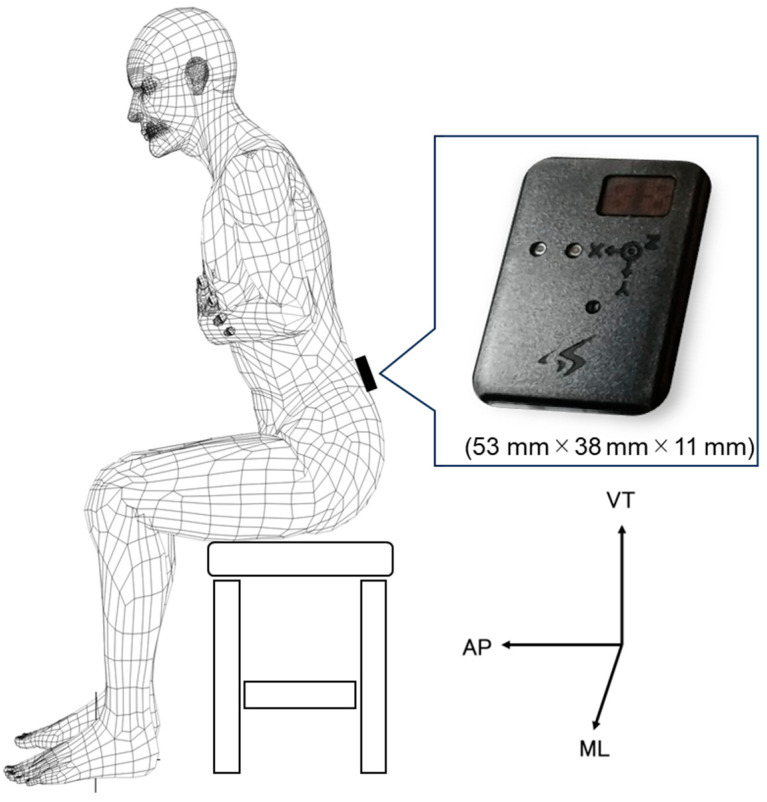
Sensor attachment position and size.

**Figure 3 geriatrics-08-00098-f003:**
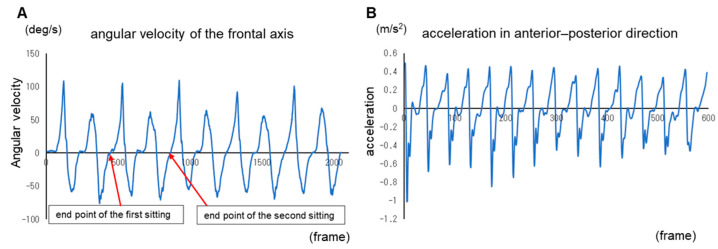
Motion signal. Specifically, sit-to-stand-to-sit motion definitions (**A**), maximum speed gait acceleration (**B**).

**Figure 4 geriatrics-08-00098-f004:**
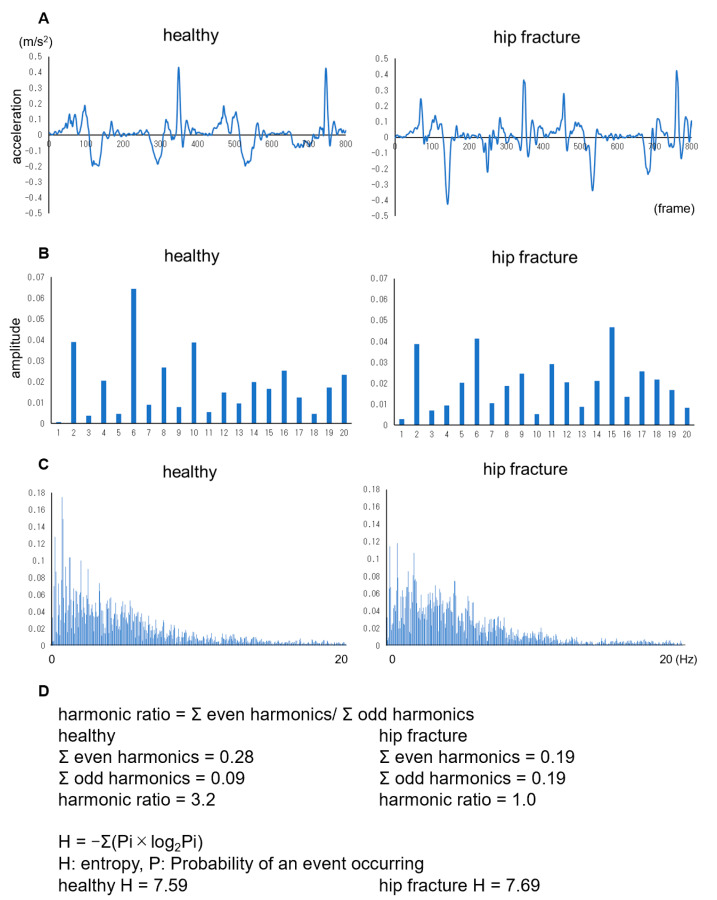
Harmonic ratio and power spectrum entropy. Specifically, vertical acceleration of two sit-to-stand-to-stand-sit of healthy and hip fracture (**A**), amplitudes of the first twenty harmonics of healthy and hip fracture (**B**), normalized probability for entropy of healthy and hip fracture (**C**), calculate of harmonic ratio and entropy of healthy and hip fracture (**D**).

**Table 1 geriatrics-08-00098-t001:** Comparison of the baseline data among the participants.

Characteristics	Healthy Group (*n* = 18)	Hip Fracture Group (*n* = 18)	*p*-Value
**Height (cm)**	154.1 ± 4.8	150.7 ± 5.5	0.055
**Age (years)**	63.1 ± 3.2	76.2 ± 7.8	<0.001
**Weight (kg)**	52.6 ± 9.0	45.3 ± 5.7	0.006
**BMI (kg/m^2^)**	22.1 ± 3.1	20.0 ± 2.4	0.029
**MMSE (points)**		27.3 ± 2.3	
**Type of hip fracture**			
Trochanteric fracture after OLIF		7 (39%)	
Subtrochanteric fracture after OLIF		2 (11%)	
Neck fracture		9 (50%)	
**Type of surgery**			
OLIF		3 (33%)	
BHA		3 (33%)	
THA		3 (33%)	
**Laterality**			
Right		7 (39%)	
Left		11 (61%)	
**Gait aid**			
None	18 (100%)	9 (50%)	
Cane	0 (0%)	9 (50%)	
**Date of measurement (days after surgery)**		46.7 ± 10.7	
**Seat height (cm)**	38.4 ± 0.8	38.9 ± 1.1	0.10

Mean ± standard deviation. BMI: body mass index, MMSE: mini-mental state examination, OLIF: open reduction internal fixation, BHA: bipolar hemiarthroplasty, THA: total hip arthroplasty. Data are expressed as number (%).

**Table 2 geriatrics-08-00098-t002:** Comparison of 5STS and MGS between the hip fracture and healthy groups.

	Healthy Group	Hip Fracture Group	*p*-Value ^†^	Effect Size, d	*p*-Value ^‡^
5STS (s)	8.75 ± 1.05	12.17 ± 3.20	<0.001	1.43	0.754
MGS (m/s)	1.93 ± 0.21	1.09 ± 0.27	<0.001	3.51	0.012

Mean ± standard deviation. 5STS: five sit-to-stand test, MGS: maximum gait speed. ^†^: Comparison between the two groups, ^‡^: analysis of covariance.

**Table 3 geriatrics-08-00098-t003:** Comparison of HR estimated for STSF and STS2s motions.

HR				
	Healthy Group	Hip Fracture Group	*p*-Value	Effect Size, d
STSF				
ML	1.88 ± 0.72	1.56 ± 0.26	0.092	0.59
VT	3.65 ± 1.74	2.67 ± 1.01	0.047	0.69
AP	3.26 ± 1.33	2.76 ± 0.88	0.443	0.44
Triaxial	2.53 ± 1.13	2.04 ± 0.63	0.406	0.53
STS2s				
ML	2.18 ± 0.89	1.75 ± 0.74	0.125	0.52
VT	3.42 ± 1.54	2.58 ± 0.83	0.049	0.69
AP	2.68 ± 0.67	2.34 ± 0.72	0.150	0.49
Triaxial	2.69 ± 1.19	2.18 ± 0.73	0.131	0.52

Mean ± standard deviation. HR: harmonic ratio, STSF: sit-to-stand-to-sit fast, STS2s: sit-to-stand and stand-to-sit for 2 s, ML: mediolateral, VT: vertical, AP: anterior–posterior.

**Table 4 geriatrics-08-00098-t004:** Comparison of PSE estimated for STSF and STS2s motions.

PSE				
	Healthy Group	Hip Fracture Group	*p*-Value	Effect Size, d
STSF				
ML	6.01 ± 0.22	5.94 ± 0.16	0.297	0.35
VT	5.78 ± 0.24	5.68 ± 0.16	0.148	0.49
AP	5.78 ± 0.16	5.61 ± 0.16	0.002	1.10
Triaxial	5.85 ± 0.21	5.68 ± 0.18	0.015	0.85
STS2s				
ML	7.63 ± 0.31	7.82 ± 0.16	0.035	0.74
VT	7.46 ± 0.22	7.63 ± 0.16	0.010	0.88
AP	7.47 ± 0.15	7.61 ± 0.17	0.014	0.86
Triaxial	7.45 ± 0.25	7.66 ± 0.17	0.003	0.97

Mean ± standard deviation. PSE: power spectrum entropy, STSF: sit-to-stand-to-sit fast STS2s: sit-to-stand and stand-to-sit for 2 s, ML: mediolateral, VT: vertical, AP: anterior–posterior.

## Data Availability

Not applicable.

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
