# Peer review of "Smoothness Evaluation Indices during Sit-to-Stand-to-Sit Motions in Healthy Older Females and after Hip Fracture Using an Accelerometer: A Pilot Study"

_geriatrics, 2023, doi:10.3390/geriatrics8050098_

Round 1
Reviewer 1 Report
Notwithstanding the relevance and relevance of the topic, I would like to make some comments about the manuscript.
Title and abstract
The authors do not indicate the study’s design with Title and abstract 1 a commonly used term in the title or the abstract. Perhaps your study will be a case-control. With such a small sample, the study should be indicated as a pilot study in the title
Introduction
In the sentence on lines 26-27, cite some studies or a systematic review.
Even if the abbreviations are never cited in the abstract, open the text again (PSE)
Methods
Organize the method please following the topics: study, Samples caracteristics, Describe the setting, locations, and relevant dates, including periods of recruitment (Explain how the study size was arrived at), exposure, follow-up, and data collection procedures, ethical statement (number or code of ethical committee document.
a) In the case of Case-control study: Give the eligibility criteria, and the sources and methods of
case ascertainment and control selection. Give the rationale for the choice of cases
and controls
b) In the case of Cross-sectional study: Give the eligibility criteria, and the sources and methods of selection of participants
Results
Put table 1 (characterization of the sample and comparison of two study subgroups) as the first to present our results.
Do not roll the tables, the abbreviations fear to appear open again.
Try to present two results in the graph, this may attract the attention of the reader.
Statistics
In the case of Case-control study—If applicable, explain how matching of cases and controls was
addressed
In the caso of Cross-sectional study—If applicable, describe analytical methods taking account of
sampling strategy
Discussion
After describe key results, please open a new topics and describe: i) liminations and strenghs, ii) interpretantion: Give a cautious overall interpretation of results considering objectives, limitations, multiplicity of analyses, results from similar studies, and other relevant evidence; iii) generalizability; iv) directions for futures studies, v) pratical implications
Author Response
Response to Reviewer 1 Comments
Thank you for your thoughtful and constructive feedback on our manuscript entitled “Smoothness evaluation indices during sit-to-stand-to-sit motions in healthy older women and after hip fracture using an accelerometer: A pilot study.” We agree with your feedback and have revised our manuscript in accordance with the content rules of Geriatrics. The corrections are numbered and added in red.
Reviewer 1
Comments and Suggestions for Authors
Title and abstract
The authors do not indicate the study’s design with Title and abstract 1 a commonly used term in the title or the abstract. Perhaps your study will be a case-control. With such a small sample, the study should be indicated as a pilot study in the title
(P1, Lines 4. P1, Lines 13–14.):
Thank you for your suggestion. We changed the title to “Smoothness evaluation indices during sit-to-stand-to-sit motions in healthy older women and after hip fracture using an accelerometer: A pilot study” and added “cross-sectional study” to the Methods section.
Introduction
In the sentence on lines 26-27, cite some studies or a systematic review.
(P1, Lines 27.):
Thank you for your suggestion; we have added the review article.
Even if the abbreviations are never cited in the abstract, open the text again (PSE)
Thank you for providing these insights. We have used the term “power spectrum entropy (PSE)” in the Abstract and Background section.
Methods
Organize the method please following the topics: study, Samples caracteristics, Describe the setting, locations, and relevant dates, including periods of recruitment (Explain how the study size was arrived at), exposure, follow-up, and data collection procedures, ethical statement (number or code of ethical committee document.
- a) In the case of Case-control study: Give the eligibility criteria, and the sources and methods of
case ascertainment and control selection. Give the rationale for the choice of cases
and controls
- b) In the case of Cross-sectional study: Give the eligibility criteria, and the sources and methods of selection of participants
(P2, Lines 63–67.):
Thank you for your suggestion; we have corrected the inadequacies according to your advice.
Results
Put table 1 (characterization of the sample and comparison of two study subgroups) as the first to present our results.
Do not roll the tables, the abbreviations fear to appear open again.
Try to present two results in the graph, this may attract the attention of the reader.
(P7, Lines 149–153. Table 1. Figure 4.):
Thank you for providing these insights. Table 1 is shown at the beginning of the Results section, and some graphs of the results have been added.
Statistics
In the case of Case-control study—If applicable, explain how matching of cases and controls was
addressed
In the caso of Cross-sectional study—If applicable, describe analytical methods taking account of
sampling strategy
(P3, Lines 122–133.):
Thank you for your suggestion. We have corrected the inadequacies of the cross-sectional study.
Discussion
After describe key results, please open a new topics and describe: i) liminations and strenghs, ii) interpretantion: Give a cautious overall interpretation of results considering objectives, limitations, multiplicity of analyses, results from similar studies, and other relevant evidence; iii) generalizability; iv) directions for futures studies, v) pratical implications
(P12, Lines 258–268.):
Thank you for your suggestions. We have added and modified the manuscript. Please verify the revision.
Reviewer 2 Report
The reviewed manuscript is devoted to the analysis of the biomechanics of movements in elderly patients. In this study, two groups were compared with and without hip fractures.
It is important to note that the authors analyzed not only the main characteristics of movement: the maximum gait speed (MGS), exercise 5STS speed (time for five sit to stand test), but also such an important characteristic of movement as smoothness. The smoothness of movement was characterized by indicators of the spectral analysis of movement: harmonic ratio (HR) power spectrum entropy (PSE) of the accelerometer sensor signal.
The section "Materials and Methods" provides information about the contingent of patients and the methods used. It is gratifying that the authors correctly planned the design of the study, including the calculation of the size of the groups.
In the chapter "results" the obtained data are presented in detail and in the required volume.
In the chapter "discussion" the analysis of the received data is carried out. Based on this, a correct conclusion was made.
The work as a whole makes a good impression.
1) Please provide here and continue to provide graphic illustrative material
- photo of the patient with accelerometer
- the given spectra of the accelerometer signal for two patients from different groups.
- complex graphic display of HR and PSD values. For example, by the radial diagrams.
2) the discussion should include a more detailed description of the biomechanical equivalent and psychophysiological reasons for the differences identified. For example, patients' loss of confidence when performing a task. Changing the biomechanical strategy for achieving results. Or similar ones.
These comments are advisory in nature.
3) It should be noted that MGS and 5STS demonstrate a relative difference in values between groups (No fracture / fracture) of 43% and 39%, respectively. I made estimates based on your data. The average coefficient of variation in the groups is 18% and 16%. Meanwhile, the HR and PSE parameters studied in the work have a relative difference in values between the groups (No fracture / fracture) of 18% and 2.8%, respectively. At the same time, the average coefficient of variation in the groups is quite large compared to the relative difference of 38% and 2.75%. This means that the simple movement characteristics MGS and 5STS have the advantage of more reliable separation of the Fr+/Fr- groups compared to the more complex HR and PSE scores. This remark does not require amendments to the publication, but is important for further evaluation of the prospects for applying these approaches.
Important remarks (please correct text):
In «Discussion»
The delay perspective of the work is to determine the objective indicators of background physical activity outside the performance of targets. This will help plan a more comfortable and effective rehabilitation strategy for patients.
In «Conclusion»
Do not use abbreviations in the conclusion.
Author Response
Response to Reviewer 2 Comments
Thank you for your thoughtful and constructive feedback on our manuscript entitled “Smoothness evaluation indices during sit-to-stand-to-sit motions in healthy older women and after hip fracture using an accelerometer: A pilot study.” We agree with your feedback and have revised our manuscript in accordance with the content rules of Geriatrics. The corrections are numbered and added in red.
Reviewer 2
The reviewed manuscript is devoted to the analysis of the biomechanics of movements in elderly patients. In this study, two groups were compared with and without hip fractures.
It is important to note that the authors analyzed not only the main characteristics of movement: the maximum gait speed (MGS), exercise 5STS speed (time for five sit to stand test), but also such an important characteristic of movement as smoothness. The smoothness of movement was characterized by indicators of the spectral analysis of movement: harmonic ratio (HR) power spectrum entropy (PSE) of the accelerometer sensor signal.
Thank you for your comment.
The section "Materials and Methods" provides information about the contingent of patients and the methods used. It is gratifying that the authors correctly planned the design of the study, including the calculation of the size of the groups.
Thank you for your confirmation.
In the chapter "results" the obtained data are presented in detail and in the required volume.
In the chapter "discussion" the analysis of the received data is carried out. Based on this, a correct conclusion was made.
Thank you for your comment.
The work as a whole makes a good impression.
1) Please provide here and continue to provide graphic illustrative material
- photo of the patient with accelerometer
- the given spectra of the accelerometer signal for two patients from different groups.
- complex graphic display of HR and PSD values. For example, by the radial diagrams.
(Figure 2. Figure 4.):
Thank you for your suggestion. We have added a figure. Please verify the revision.
2) the discussion should include a more detailed description of the biomechanical equivalent and psychophysiological reasons for the differences identified. For example, patients' loss of confidence when performing a task. Changing the biomechanical strategy for achieving results. Or similar ones.
These comments are advisory in nature.
(P12, Lines 263–265.):
Thank you for your suggestion. We have added the indicated information to the limitations.
3) It should be noted that MGS and 5STS demonstrate a relative difference in values between groups (No fracture / fracture) of 43% and 39%, respectively. I made estimates based on your data. The average coefficient of variation in the groups is 18% and 16%. Meanwhile, the HR and PSE parameters studied in the work have a relative difference in values between the groups (No fracture / fracture) of 18% and 2.8%, respectively. At the same time, the average coefficient of variation in the groups is quite large compared to the relative difference of 38% and 2.75%. This means that the simple movement characteristics MGS and 5STS have the advantage of more reliable separation of the Fr+/Fr- groups compared to the more complex HR and PSE scores. This remark does not require amendments to the publication, but is important for further evaluation of the prospects for applying these approaches.
Thank you for providing these insights.
Important remarks (please correct text):
In Discussion
The delay perspective of the work is to determine the objective indicators of background physical activity outside the performance of targets. This will help plan a more comfortable and effective rehabilitation strategy for patients.
(P12, Lines 266–268.):
Thank you for your suggestion. We have added information on the indicators that should be considered for planning effective rehabilitation in the future. Please check the correction.
In Conclusion
Do not use abbreviations in the conclusion.
(P12, Lines 270–278.):
Thank you for providing these insights. Please verify the revision.
Reviewer 3 Report
This manuscript examines quality indicators (i.e., harmonic ratio, power spectrum entropy) in terms of smoothness in sit-to-stand movements between female healthy group and hip fracture group. The topic of the study is interesting and the study results might have a potential clinical impact, but the manuscript lacks important information, which should be addressed by considering the comments and suggestions below: Major comments: Abstract: This section needs the most revision by providing more information about this study and p-value(s). Introduction: This section needs the most revision by providing reasonable rationales and evidence. Please describe why the authors chose 5-times sit-to-stand as fast as possible (STSF) and 2-second for each motion (STS2s) in female healthy group and hip fracture group. The authors need more focus on why the use of an accelerometer for STS analysis would be required for this population and previous finding, for example, significant differences in other accelerometer variables between healthy and hip fracture groups. Also, the authors should clearly describe gaps between existing knowledge/evidence/practice and their work. English needs to be improved extensively. Please add hypotheses. Methods: Overall, this section needs the most revision as it reads unclearly. The authors should provide flow chart of study population. Please describe why the authors chose females as a targeting population for this study? Please provide the motion sensor size. If possible, provide a figure with a subject wearing the motion sensor for better understanding. Please provide the rationale why the authors measure the seat height. Table 1, provide the % for the categorical variables. This reviewer strongly recommends that the authors include a Figure showing the STSF and STS2s sensor signals of the both healthy and hip fracture groups. Then, the readers can clearly understand the differences of the smoothness from the accelerometer. Please describe how the authors defines the movement starts and ends for the signal processing. This reviewer strongly recommends that the authors provides a figure for HR and PSE of the both healthy and hip fracture groups. Then, the readers can clearly understand the differences of the smoothness. Page 3, Line 112-114: The authors mentioned that “STSF analyzed 512 intermediate movements (5.12 s), whereas STS2s analyzed 2048 intermediate movements (20.48 s)”. This reviewer thinks that this sentence should go to the Results section, not Methods. Page 3, Line 118-120: The authors mentioned that “Effect sizes were calculated in terms of (r), where r=0.1, r=0.3, and r=0.5 represented small, medium, and large effect sizes, respectively”. Please add a reference for this. In addition, please describe why the authors use the Pearson’s r for effect size, instead of the other type of effect size. Results: During performing STS2s using metronome, this reviewer thinks that especially hip fracture group may not perform STS within 2 sec. If yes, how did you analyze it? The authors provide Table 2, 3 and 4 in this Results section. Without figure(s) representing the accelerometer signals between healthy and hip fracture groups for Table 2 (motion signals of STS movement time, max. gait speed with start and end points). To interpret the HR and PSE values, since there is no equation and explanation, the readers may not understand what high and low values in HR and PSE indicates. Please clarify it. Most importantly, all of the results were not adjusted the age and BMI. This reviewer knew that the authors mentioned it as a limitation. However, it is not acceptable to report the results without adjusting age and BMI since age associated with frailty, various disease, etc. is very important variable to evaluate the physical movements. Hip fracture group’s age were significantly higher than healthy. It may be because of older age, not because of hip fracture. Thus, this reviewer strongly recommends for adjusting the age and BMI variables statistically. Discussion The authors mentioned that “In previous studies on healthy participants by age, it has been reported that the average time taken for 5STS was 7.8, 9.3, and 10.8 s for participants in their 60s, 70s, and 80s, the required time tended to be longer in older age groups.” What about this study’s 5STS in the 60s, 70s, and 80s? How many subjects were for each age categories (60s, 70s, and 80s)? This section needs the most revision by considering the major comments and questions above. Please address the comments clearly and comprehensively. English needs to be improved extensively.Author Response
Response to Reviewer 3 Comments
Thank you for your thoughtful and constructive feedback on our manuscript entitled “Smoothness evaluation indices during sit-to-stand-to-sit motions in healthy older women and after hip fracture using an accelerometer: A pilot study.” We agree with your feedback and have revised our manuscript in accordance with the content rules of Geriatrics. The corrections are numbered and added in red.
Reviewer 3
Comments and Suggestions for Authors
This manuscript examines quality indicators (i.e., harmonic ratio, power spectrum entropy) in terms of smoothness in sit-to-stand movements between female healthy group and hip fracture group. The topic of the study is interesting and the study results might have a potential clinical impact, but the manuscript lacks important information, which should be addressed by considering the comments and suggestions below: Major comments: Abstract: This section needs the most revision by providing more information about this study and p-value(s).
Thank you for your comment.
Introduction: This section needs the most revision by providing reasonable rationales and evidence. Please describe why the authors chose 5-times sit-to-stand as fast as possible (STSF) and 2-second for each motion (STS2s) in female healthy group and hip fracture group.
Thank you for your comments. We selected STSF and STS2s to confirm differences in movement speed. Notably, STSF was selected because it has been reported in many previous studies, is easy to measure, and is less burdensome for the patient. If useful evaluation indices can be analyzed using 5STS, the possibility of clinical application may be expanded. The reason for selecting STS2s was to evaluate 1- and 3-second movements in preliminary experiments. However, 1-second movements were too fast for older adults and patients with disabilities, and 3-second movements were too slow; therefore, 2-second STS2s was selected owing to the low feasibility of evaluation.
The authors need more focus on why the use of an accelerometer for STS analysis would be required for this population and previous finding, for example, significant differences in other accelerometer variables between healthy and hip fracture groups. Also, the authors should clearly describe gaps between existing knowledge/evidence/practice and their work. English needs to be improved extensively. Please add hypotheses.
(P2, Lines 52–54. P2, Lines 58–60.):
Thank you for providing these insights. The reason for using accelerometers for STS analysis is that most current evaluations of standing movements are based on observation and movement times, and we believe that it necessary to evaluate the quality of movements. Some studies have evaluated sit-to-stand motions using accelerometers in the older healthy and hip fracture groups in terms of movement time, mobility time, and quantity of physical activity; however, to the best of our knowledge, none of these studies have evaluated the quality of sit-to-stand motion. In clinical settings, some people with fast movements might have inadequate quality of movement. Our study hypothesized that the healthy older group will have higher HR and lower PSE because the quality of movements is expected to be higher in the healthy older group than in the hip fracture group during the hospital discharge preparation period when there is a high possibility of residual instability in sit-to-stand-to-sit motions. Please verify the revision.
Methods: Overall, this section needs the most revision as it reads unclearly. The authors should provide flow chart of study population.
(P2, Lines 63–67. Figure 1.):
Thank you for your suggestion. We have added the content of the prior explanation and mobilization for the participants to whom it was adapted.
Please describe why the authors chose females as a targeting population for this study?
Thank you for providing these insights. We only included female respondents owing to the difficulty in recruiting male respondents. We will need to collect more data in the future.
Please provide the motion sensor size. If possible, provide a figure with a subject wearing the motion sensor for better understanding.
(P2, Lines 73–74. Figure 2.):
Thank you for your suggestion. Please check the figure and the size of the sensor.
Please provide the rationale why the authors measure the seat height.
Thank you for providing these insights. We are presenting the results of this study because we adjusted the seat height to compare the sit-stand-to-sit motion abilities, excluding differences in physical conditions as much as possible.
Table 1, provide the % for the categorical variables.
(Table 1.):
Thank you for your suggestion. Please check the correction.
This reviewer strongly recommends that the authors include a Figure showing the STSF and STS2s sensor signals of the both healthy and hip fracture groups. Then, the readers can clearly understand the differences of the smoothness from the accelerometer. Please describe how the authors defines the movement starts and ends for the signal processing. This reviewer strongly recommends that the authors provides a figure for HR and PSE of the both healthy and hip fracture groups. Then, the readers can clearly understand the differences of the smoothness.
(P3, Lines 98–106. Figure 3. Figure 4):
Thank you for your suggestion. The explanation was insufficient. We have added a figure and explanation. Please check the correction.
Page 3, Line 112-114: The authors mentioned that “STSF analyzed 512 intermediate movements (5.12 s), whereas STS2s analyzed 2048 intermediate movements (20.48 s)”. This reviewer thinks that this sentence should go to the Results section, not Methods.
(P7, Lines 168–169.):
Thank you for your suggestion. We have mentioned it in the Results section. Please check the correction.
Page 3, Line 118-120: The authors mentioned that “Effect sizes were calculated in terms of (r), where r=0.1, r=0.3, and r=0.5 represented small, medium, and large effect sizes, respectively”. Please add a reference for this. In addition, please describe why the authors use the Pearson’s r for effect size, instead of the other type of effect size.
(P3, Lines 124–126.):
Thank you for the suggestion. Regarding the comparison without correspondence, we think you have pointed out that the appropriate effect size is d. There were some tests that were not normally distributed; therefore, we used r to make them consistent. Please check the reference.
Results: During performing STS2s using metronome, this reviewer thinks that especially hip fracture group may not perform STS within 2 sec. If yes, how did you analyze it?
(P2, Lines 63–67.):
Thank you for your suggestion. Impossibility of motion was excluded. The explanation was insufficient; therefore, it has been corrected.
The authors provide Table 2, 3 and 4 in this Results section. Without figure(s) representing the accelerometer signals between healthy and hip fracture groups for Table 2 (motion signals of STS movement time, max. gait speed with start and end points).
(Figure 3. Figure 4.):
Thank you for your suggestion. Please check the correction.
To interpret the HR and PSE values, since there is no equation and explanation, the readers may not understand what high and low values in HR and PSE indicates.
(Figure 4.):
Thank you for providing these insights. We have added a figure based on a previous study. Please check the correction.
Please clarify it. Most importantly, all of the results were not adjusted the age and BMI. This reviewer knew that the authors mentioned it as a limitation. However, it is not acceptable to report the results without adjusting age and BMI since age associated with frailty, various disease, etc. is very important variable to evaluate the physical movements. Hip fracture group’s age were significantly higher than healthy. It may be because of older age, not because of hip fracture. Thus, this reviewer strongly recommends for adjusting the age and BMI variables statistically.
(P7, Lines 158–160. Table 2.):
Thank you for your suggestion; accordingly, we have added the results of the analysis of covariance for 5STS and maximal gait speed. After adjusting for the covariates of age and BMI, maximum gait speed was significantly higher in the healthy group than in the hip fracture group, and there was no significant difference in 5STS. We believe that quality indicators may provide different information than movement time. Please check the added and revised data.
Discussion: The authors mentioned that “In previous studies on healthy participants by age, it has been reported that the average time taken for 5STS was 7.8, 9.3, and 10.8 s for participants in their 60s, 70s, and 80s, the required time tended to be longer in older age groups.” What about this study’s 5STS in the 60s, 70s, and 80s? How many subjects were for each age categories (60s, 70s, and 80s)? This section needs the most revision by considering the major comments and questions above. Please address the comments clearly and comprehensively.
Comments on the Quality of English Language
English needs to be improved extensively.
(P11, Lines 214–218.):
Thank you for providing these insights. The 5STS of 17 patients in their 60s in the healthy group was 8.8 ± 1.1 seconds, and that of 3 patients in their 60s, 9 in their 70s, and 5 in their 80s in the hip fracture group was 9.7 ± 0.9, 12.0 ± 2.9, and 12.5 ± 2.8 seconds, respectively. Please check the correction.
Reviewer 4 Report
GERIATRICS-2515665 presents results for comparisons of STS motions in healthy and hip fracture older females. While some parts of this paper were interesting, other areas could be improved. I hope the authors consider my feedback.
MAJOR COMMENTS
· Lines 79-80: Gait speed tends to be at comfortable speed. Please provide support in the text (e.g., citation) for fast gait speed.
· Table 1 is out of place and should be moved to the Results.
· Table 1: Just list the p-value. No need for all the “*”. The same applies to other tables.
· Section 2.3: Remove r-values for effect sizes, as this may confuse the reader with correlation coefficients. Use something else of just spell out “effect size” where appropriate.
· Results text should include the mean+-SD difference of group comparisons, not just p-values and effect sizes.
MINOR COMMENTS
· Lines 41-51: The sentences here tend to run-on and repeat in sort. Consider revision for flow and presentation.
· Limitations: The use of 20 Hz is defended with a citation, but higher frequency may have improved the resolution of these data.
· Line 101: Avoid abbreviations for all headers.
· Lines 122-125 should go to the beginning of the paragraph.
· Expand the limitations paragraph a bit more.
· Make any changes to the abstract that align with those from the text.
Author Response
Response to Reviewer 4 Comments
Thank you for your thoughtful and constructive feedback on our manuscript entitled “Smoothness evaluation indices during sit-to-stand-to-sit motions in healthy older women and after hip fracture using an accelerometer: A pilot study.” We agree with your feedback and have revised our manuscript in accordance with the content rules of Geriatrics. The corrections are numbered and added in red.
Reviewer 4
Comments and Suggestions for Authors
GERIATRICS-2515665 presents results for comparisons of STS motions in healthy and hip fracture older females. While some parts of this paper were interesting, other areas could be improved. I hope the authors consider my feedback.
Thank you for your comment.
MAJOR COMMENTS
Lines 79-80: Gait speed tends to be at comfortable speed. Please provide support in the text (e.g., citation) for fast gait speed.
(P2, Lines 89–90.):
Thank you for your suggestion. Please check the revised manuscript and reference citations.
Table 1 is out of place and should be moved to the Results.
(P7, Lines 149–153.):
Thank you for your suggestion. The table has been moved to the Results section. Please check the correction.
Table 1: Just list the p-value. No need for all the “*”. The same applies to other tables.
(Table 1–4.):
Thank you for your suggestion; it has been removed. Please check the correction.
Section 2.3: Remove r-values for effect sizes, as this may confuse the reader with correlation coefficients. Use something else of just spell out “effect size” where appropriate.
(P9, Lines 124–126. Table 2–4):
Thank you for providing these insights. Please check the effect size, which has been changed from r to d and corrected.
Results text should include the mean+-SD difference of group comparisons, not just p-values and effect sizes.
(P9, Lines 154–176.):
Thank you for your suggestion. Please check the added text.
MINOR COMMENTS
Lines 41-51: The sentences here tend to run-on and repeat in sort. Consider revision for flow and presentation.
(P2, Lines 52–59.):
Thank you for providing these insights. Please check the correction.
Limitations: The use of 20 Hz is defended with a citation, but higher frequency may have improved the resolution of these data.
(P12, Lines 262–263.):
Thank you for your suggestion. Additions were made to the limitations. Please check the correction.
Line 101: Avoid abbreviations for all headers.
Lines 122-125 should go to the beginning of the paragraph.
Expand the limitations paragraph a bit more.
Make any changes to the abstract that align with those from the text.
(P7, Lines 154,161,167. P7, Lines 149–253. P12, Lines 258–268.):
Thank you for providing these insights. Please check the corrected version.
Round 2
Reviewer 1 Report
Dear authors, thank you for your effort. The manuscript has undergone substantial increments. Information was added to the image and graphic output about the technological system used for movement monitoring. This makes the manuscript very attractive to the reader, as it reveals new information. However, I still want to ask for some small changes:
1) In the section methods: a)please place the information separately and for this, open subsections: 2.1 Study design; 2.2 Particpants (including the way of capturing the sample... convenient, I guess?), 2.3 inclusion and exclusion criteria, 2.3 Ethical procedures, 2.4. Assessments; 2.4.1 - ....
2) In the section discussion: please place the following information: 4. Discussions, 4.1 - Strenghs and limitations; 4.2 Perspective for futures studies; 4.3 Practical implications
Thank you.
Reviewer 3 Report
From my point of view, the manuscript is now ready to be published.
The final check may be needed.
Reviewer 4 Report
The authors did a nice job addressing my previous concerns. However, please insert some actual statistics to support the statements related to the results in the abstract.
